# A Subspace Pre-Learning Strategy to Break the Interpose PUF

Gaoxiang Li [1,*] and Khalid T. Mursi [2]

1. Department of Computer Science, Texas Tech University, Lubbock, TX 79709, USA
2. Department of Cybersecurity, College of Computer Science and Engineering, University of Jeddah, Jeddah 21959, Saudi Arabia; kmursi@uj.edu.sa
* Correspondence: gaoli@ttu.edu

**Abstract:** Physical Unclonable Functions (PUFs) are promising security primitives for resource-constrained IoT devices. A critical aspect of PUF security research is to identify all potential security risks. This information about vulnerabilities is beneficial for both PUF developers and PUF-using application developers in terms of designing new PUFs to mitigate existing risks and avoid vulnerable PUFs. Recently, a PUF structure called Interpose PUF (IPUF) was proposed, which claims to be resistant to reliability attacks and machine learning modeling attacks. Related studies on this secure PUF design have demonstrated that some IPUFs can still be broken, but large IPUFs may remain secure against all known modeling attacks. In addition, all these studies either focus on plain challenge–response pair attacks or require prior knowledge of IPUF architecture implementation. However, depending on the claim of attack resistance to reliability attacks, we can employ a different attack approach to break IPUFs. In this paper, we describe a subspace pre-learning-based attack method that can rapidly and accurately break the IPUFs that were treated as secure in the earlier study, revealing a vulnerability in IPUFs if the open interface conforms to the way challenge–response data are accessed by the subspace pre-learning-based attack method.

**Keywords:** IoT security; physical unclonable function; interpose PUF; machine learning modeling attack

## 1. Introduction

The Internet of Things (IoT) has wide and deep participation in business and everyday life. With the exponential rise of IoT requirements, communication security has attracted increased attention [1]. However, considering most traditional cryptographic techniques, which require persistent memory to achieve the desired level of security, many IoT devices are resource-constrained and cannot support traditional cryptographic protocols [2,3]. Physical Unclonable Functions (PUFs) were proposed as a potential replacement for classical cryptography in IoT devices [4–6], leveraging small physical variations of a small number of transistors to produce responses unique to the individual circuit. Because of their low resource requirements, PUFs are excellent candidates for hardware primitives that can be utilized to construct security protocols on network nodes with limited resources.

However, before adopting PUFs as a trusted security function, they must be examined to identify all possible security vulnerabilities, such as vulnerabilities to machine learning (ML) modeling attacks [7–10] and reliability-based attacks [11,12]. In machine learning modeling attacks, the attacker eavesdrops on-air packets between IoTs in order to collect enough plain challenge–response pairs (CRPs) to build a model. Then, the attacker inputs the collected challenges, as features, and responses, as class labels, into an ML model targeted at having a learned model for future response prediction. In reliability-based attacks, the attacker applies pre-set challenges to the PUF with an open interface and collects specific CRPs. Furthermore, CRPs obtained through freely queried can be easily used to break PUFs by utilizing the reliability information of these CRPs.

Interpose PUF (IPUF) is proposed by Nguyen et al., [13] to mitigate the two above-mentioned classes of attack. Studies in [8,13] show IPUFs could withstand various existing

attack methods. However, IPUFs with open interfaces are the target of a recent attack method [14] that has been able to crack many IPUFs if the position of the interpose bit is known to the attacker in advance. Furthermore, a more recent study [15] found that, using an NN-based method, it is possible to break (7,7)-IPUFs with the interpose bit at any position and without an open interface. These two studies reveal the vulnerability of IPUFs, but either study requires prior knowledge of the IPUF architecture, and they do not focus exclusively on plain challenge–response pairs without leveraging open-interface data. As a result, there is considerable potential for further research into the vulnerability of IPUFs.

In this paper, we describe a subspace pre-learning method to attack IPUFs. Specifically, we first utilize pre-set and problem-tailored subspace training datasets to pre-train an NN attacking model. Then, the pre-trained model is fine-tuned by a whole space training dataset to break the target IPUF instance. Experimental results show that this subspace learning method remarkably reduces the required training CRPs and required training times compared with early studies on our tested open-interfaced IPUFs. For example, our method only needs 300 k CRPs to break (1,5)-IPUFs, while 1 m CRPs were needed in previous study; our method needs 1.5 h to break (1,7)-IPUFs, while 17 hrs were needed in previous study, etc. Additionally, this attack method is capable of breaking (1,8)-IPUFs and (8,8)-IPUFs without prior knowledge about the position of the interpose bit, while previous research could only break (1,7)-IPUFs and (7,7)-IPUFs. Although our method does not fully break the IPUF for all conceivable sizes and complexity yet, it reveals a vulnerability in IPUFs if the IPUF-embedded device has an interface that enables people to access challenge–response data.

The remainder of this paper is organized as follows: Section 2 gives a general overview of PUF mechanisms. Section 3 explains the subspace pre-learning method we implemented to break IPUFs. The experiment and its results are presented in Section 4. Finally, concluding remarks are given in Section 5.

## 2. Background Information on PUFs

In order to clarify technical discussions in later sections, we will briefly describe the mechanism of the arbiter PUF, XOR-PUF, and IPUF in this subsection.

### 2.1. The Arbiter PUFs

Figure 1 shows a case of an arbiter PUF with n bits of challenge. An n-bit arbiter PUF is made up of n stages, each with two multiplexers (MUXs). When a rising signal is given, the signal enters the arbiter PUF from stage one and splits into two signals. The two signals are routed through gates at each stage, and the propagation paths to the multiplexers at each stage are determined by the challenge bit. Finally, two signals reach the D flip-flop, which acts as an arbiter to determine whether the signal on the top path or the signal on the lower path arrives first. If the top path signal arrives first, the D flip-flop returns 1; otherwise, it returns 0.

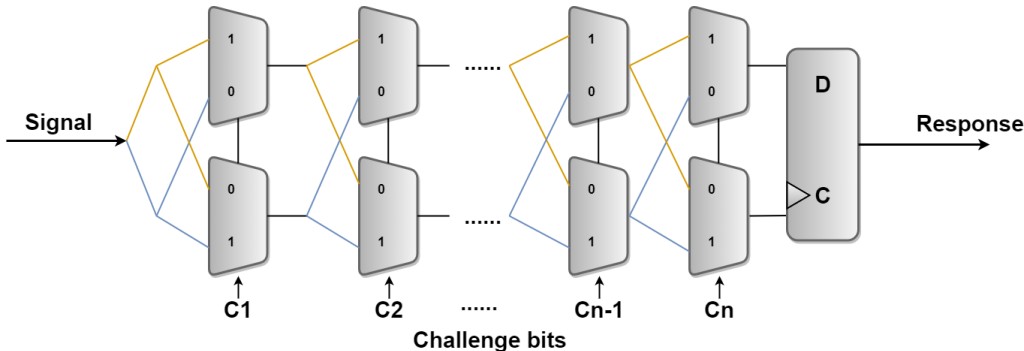

**Figure 1.** An arbiter PUF with n bits of challenge.

### 2.2. The XOR Arbiter PUFs

Due to the weak resistance of arbiter PUFs to ML modeling attacks, a new PUF was proposed in [16] that utilized a nonlinear XOR gate in conjunction with multiple arbiter PUFs to generate the final response. This type of PUF is known as the XOR arbiter PUF. The simple case of one n-bit 2-XOR-PUF is illustrated in Figure 2. A k-XOR-PUF is composed of k component arbiter PUFs (also referred to as streams or sub-challenges) in which the responses of all k component arbiter PUFs are XORed at an XOR gate to produce a single bit response.

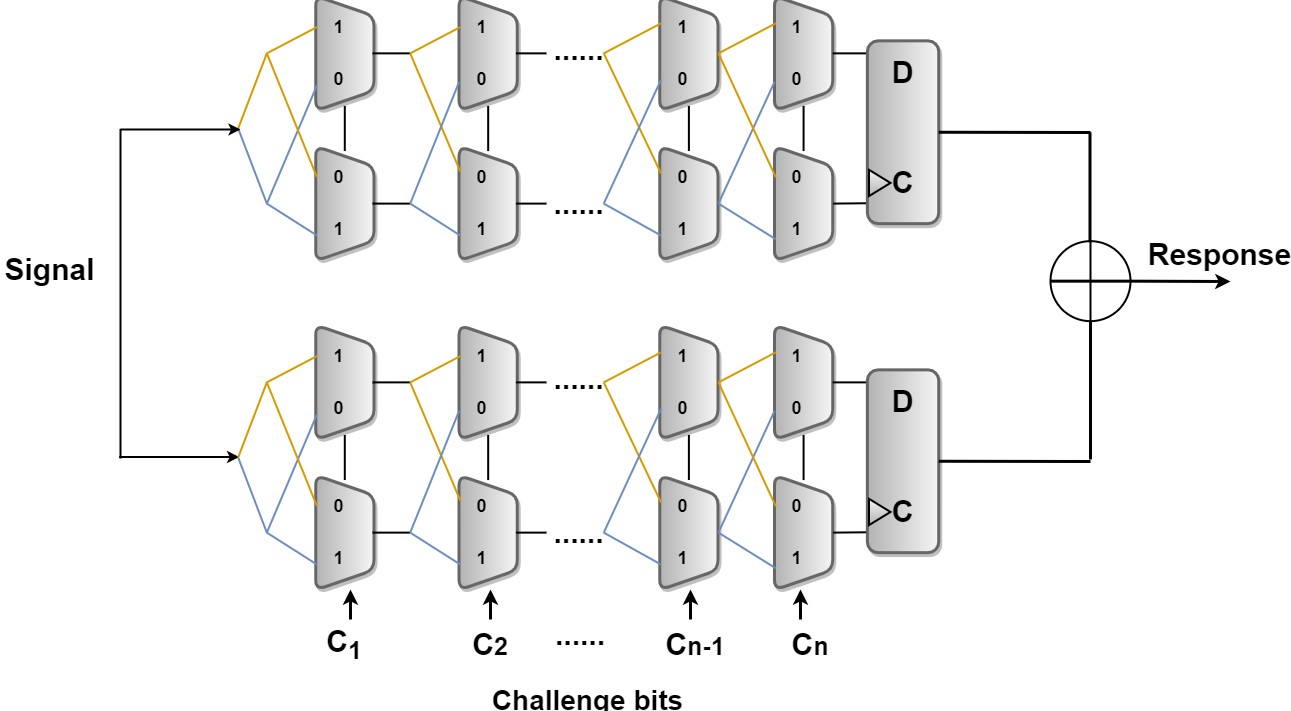

**Figure 2.** Illustration of an XOR-PUF with two arbiter PUF components and n bits of each stream; the final response is determined by the XORed result of two arbiter PUF components.

According to [17,18], XOR-PUFs are more resistant to modeling attacks than arbiter PUFs. When combined with lockdown scheme mutual authentication [2] to eliminate the open-access interface, all modeling attacks developed to date have been unable to crack the XOR-PUF within the limited number of available CRPs. Nonetheless, recent studies [19,20] demonstrated that, for 64-bit XOR-PUFs with nine or fewer component arbiter PUFs, there are attack methods capable of cracking and predicting the responses of such PUFs with a prediction accuracy of around 98 percent. Increasing the number of streams and challenge stages, on the other hand, increases the cost and power consumption of a PUF, which is a critical consideration for resource-constrained IoT devices. Additionally, as the number of streams increases, the reliability of PUFs decreases and the risk of reliability-based attacks increases [11].

### 2.3. The Interpose PUFs

An (x,y)-IPUF is constructed from an x-XOR PUF and a y-XOR PUF, with the output of the x-XOR PUF serving as a challenge bit in the y-XOR PUF. An (x,y)-IPUF is constructed by combining an $X_{up}$-XOR PUF and a $Y_{down}$-XOR PUF. By adding the interpose bit as a challenge bit to the second XOR PUF, the $Y_{down}$-XOR PUF has exactly one more stage than the $X_{up}$-XOR PUF. Figure 3 illustrates a schematic representation of IPUF. The IPUF has demonstrated to be secure against both the reliability-based attacks [11] and machine learning modeling attacks. Additional research [8] discovered that IPUFs composed of

small component XOR PUFs are vulnerable to a neural network-based attack method. Wisiol et al. [14] demonstrated that (1,9)-IPUFs or (8,8)-IPUFs can be broken when the attacker is aware of the interpose position in the IPUF. However, the attackers must obtain the information that has been interposed in this type of attack. Recent research [15] has demonstrated that, regardless of the interpose position, a more recent attack method is capable of cracking IPUFs without an open interface. This attack method is capable of breaking IPUFs with a length of (1,7) or (7,7).

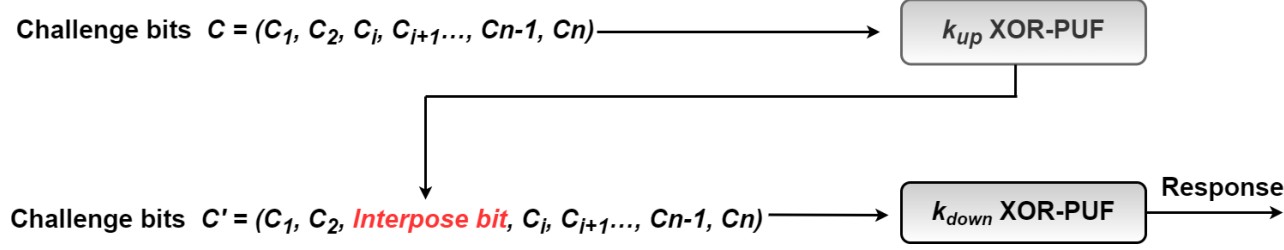

**Figure 3.** Schematic representation of Interpose PUFs.

## 3. Subspace Pre-Learning for IPUFs

Using a neural network-based method without an open interface, the (1,7)-IPUFs and (7,7)-IPUFs have been successfully broken as described in [15]. We attempted the same method with larger IPUFs, such as (1,8)-IPUFs and (8,8)-IPUFs, but were unsuccessful. When the IPUF was proposed and claimed to be immune to reliability attacks, the assumption was that IPUFs do not require a lockdown protocol [2] to prevent open interfaces from being used by reliability attacks, and freely queried CRPs can be utilized for attacking IPUFs. As a result, the attack methods against the PUF with an open interface are not limited to simple challenge–response pair attacks. For attacking the IPUF with an open interface, Wisiol et al. [14] reported a reliability-based attack method that could "split" and break IPUFs more efficiently by targeting open-interfaced IPUFs. However, this attack method requires knowing the position of the interpose bit, which is an addition to prior knowledge of the IPUF implementation. As for attacking other types of PUFs with open interfaces, Asseri et al. [12] reported success with a subspace pre-learning attack on open-interfaced component-differential challenged XOR-PUFs, which makes use of the open-interface CRPs to reduce training complexity as well as speed up the training process. Therefore, we are motivated to develop a pre-learning subspace attack approach based on the open interface IPUF.

Our subspace pre-learning attack method, which makes use of multiple sets of training data, each from a subspace with a much lower dimension than the entire challenge space, trains the PUF model restricted to each subspace using the same neural network model. As a result of this learning, the parameters of the PUF model learned in one subspace were passed as initial weight parameters for the learning of the subsequent subspace, which means that the learning results from the subspace pre-learner on the PUF are transferred onto the next subspace's pre-learner. Finally, the trained neural network weights from the pre-learning approach were input into the full space learner, which was the last step in the process. In addition, pre-training the NN model by subspace training datasets across the whole challenge space could devise rapidly estimable for the initial weights of the NN model parameters. In addition, in this way, we can speed up the attack in the final round in the whole challenge space.

In greater detail, this new pre-learning machine learning method is described in Algorithm 1. For an $(x,y)$-IPUF with k challenge bits, our proposed subspace pre-learning method includes three rounds of training. Firstly, we generate k/m training datasets and m is a pre-defined constant. There are m consecutive challenge bits that are totally randomly generated, and the rest of the k-m bits of the challenges are fixed. The positions of unfixed consecutive challenge bits are different, and there is no overlap in each subset—for example, for an IPUF with a 64-bit challenge, and we set m to 8. In the first round, we

generate eight different subspace training datasets, and the [0:7] bits in the 64-bit challenge are randomly generated in the first subspace dataset. [8:15] bits and [16:23] bits are also randomly generated for the second and third datasets, etc. The rest 56 challenge bits are fixed within subspace training datasets.

---

**Algorithm 1:** Subspace pre-learning attack method.

    **Data:** Whole space CRP set: $S$;
    Subspace CRP set in round one: $S_1$;
    Subspace CRP set in round two: $S_2$;
    NN model weight parameters: $W$;
    initialize $S, S_1, S_2$ and $W$;
    **Function** *attack_method*();
    **if** *Round $\leq$ 2* **then**
        |   *model_training*$(W, S1)$;
        |   *model_training*$(W, S2)$
    **else**
        |   *model_training*$(W, S)$
    **end**
    **Function** *model_training*$(W, S)$;
    **if** *Accuracy $\leq$ 98%* **then**
        |   Feed $S$ into NN model with weight $W$;
        |   *Accuracy = Evaluate*$(W, S)$
    **end**

---

In the second round, we generate k/m different training datasets and set the length of unfixed consecutive challenge bits to m × 2 while maintaining their different positions in each subset. The size of the dataset in the second round is larger than it was in the first round. Because the subspaces have much lower dimensionality, the model restricted to each subspace can be learned very quickly; however, the model learned on subspaces is not highly accurate in the entire space. Therefore, we need round three with the learning from the first two rounds as pre-learners for the whole space learning. An overview of the first two training rounds is given in Figures 4 and 5.

## First Training Round

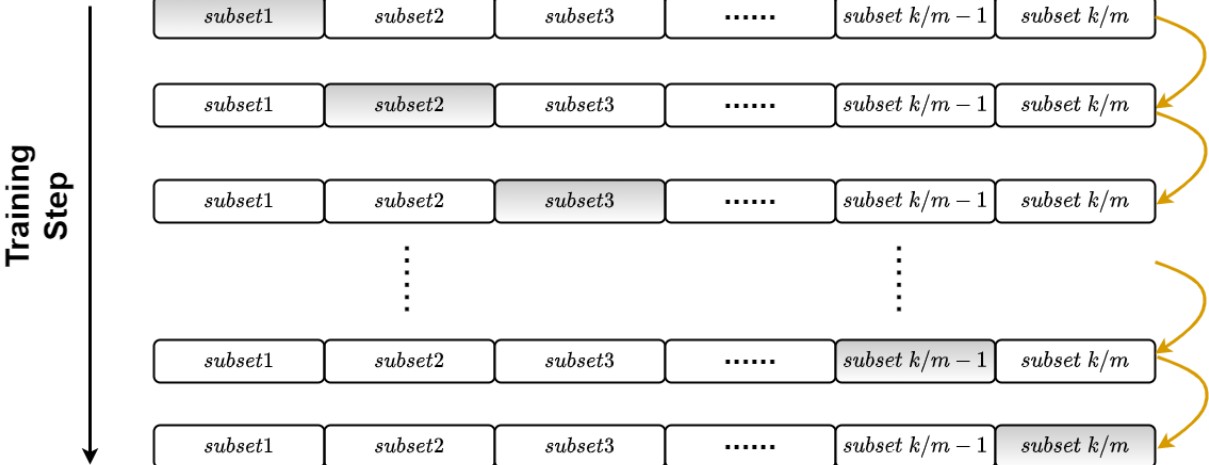

**Figure 4.** An overview of the first-round pre-learning. The challenge bits of a subset in color are randomly generated, and the rest of the bits are fixed.

## Second Training Round

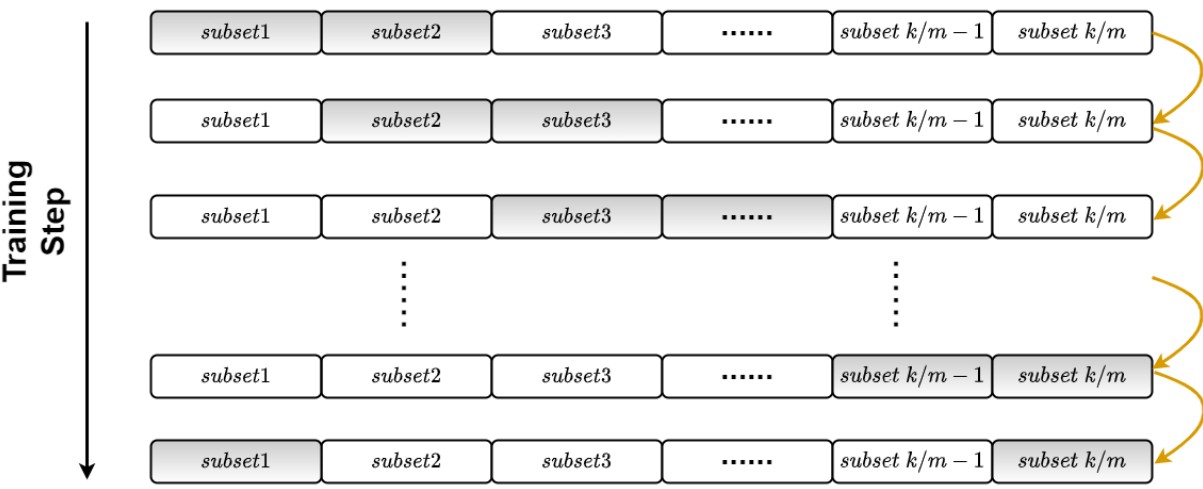

**Figure 5.** An overview of the second-round of pre-learning The challenge bits of two subsets in color are unfixed, and the rest of the bits are fixed.

## 4. Experimental Studies

### 4.1. Generating CRP from a Simulator

In the experiments, we used a simulator that is based on the Pypuf library [20]. For each IPUF architecture setting, we generate 20 different simulated 64-bit IPUF instances and 150 million CRP for each instance as the whole training space. As for the length of each training subset, for an $(x,y)$-IPUF, if $x + y \leq 9$, we assign m to 8; if $x + y > 9$, we assign m to 16. In addition, we generate 2 m CRP (up to 1 million) for the first-round subspace training dataset and $10 \times 2$ m (up to 8 million) CRP for the second-round subspace training dataset. The generated PUF instances are all from the normal distribution, with a mean of 0 and a standard deviation of 1, and no noise value was added. In addition, the interpose position is random, and this information is not used in our modeling attack method.

### 4.2. Experiment Setup

In the experiments, we chose the NN-based method used by Thapaliya et al. [15] as the baseline method as well as the base learner method for the subspace pre-learning approach. This NN-based method contains three hidden layers and uses tanh as the activation function. In addition, this baseline NN-based method could break (1,7)-IPUFs and (7,7)-PUFs with 6 million CRP (without open interface). The parameters of the NN attack method we used are exactly the same as the method used by Thapaliya et al. to evaluate the performance. For the convenience of reading, the parameters are listed in Table 1. The experiments employ a 10–90 testing–training split, with 1% CRP from the training set used for validation. The code is all implemented in Python using the TensorFlow and Keras ML libraries [21,22]. Furthermore, the maximum number of CRP generated from the simulator is 150 million; experiments stop when the testing cannot converge with the maximum number of available CRP.

**Table 1.** Parameters of the NN attack method for $(x,y)$-IPUFs.

| Parameters | Description |
| --- | --- |
| Optimizing Method | ADAM |
| Output Activation Function | Sigmoid |
| Learning Rate | Adaptive |

**Table 1.** *Cont.*

| Parameters | Description |
|---|---|
| Layer Size | Layer1 = 128<br>Layer2 = 64<br>Layer3 = 128 |
| Loss Function | Binary cross entropy |
| Batch Size | First round: 1k<br>Second round: 10k<br>Whole space: $10^{y-1}$ |
| Kernel Initializer | Random Normal |
| Early Stopping | True, when validation accuracy is 98% |

### 4.3. Experimental Results and Discussion

The experimental results of the ML modeling attack on IPUFs with the proposed subspace pre-learning approach are listed in Table 2. The results of the general NN-based modeling attack method [15] without open interface on IPUFs, and the results of reliability-based attack method [14] with known interpose bit position are also added to verify the performance of our proposed method. The "Security Evaluator" column indicates the attacking method we used for each testing result row, and only testing accuracy higher than 85% is considered a successful attack. The method "A" in the table refers to the method implemented by Thapaliya et al. [15]. The method "B" in the table refers to the method implemented by Wisiol et al. [14]. Method "C" refers to the subspace pre-learning method.

**Table 2.** Experimental results on attacking the IPUF dataset (Method "A" refers to the method in [15]. Method "B" refers to the method in [14]. Method "C" refers to the proposed subspace pre-learning method).

| Number of Stages | (*x*,*y*)-IPUF | Security Evaluator | Training Size | Average Accuracy | Training Time | Success Rate |
|---|---|---|---|---|---|---|
| 64 bits | 1,5 | A | 1 m | 99% | 2 min | 80% |
| | | B | 500 k | 95% | 9 min | 100% |
| | | C | 300 k | 99% | 1 min | 90% |
| | 1,6 | A | 2 m | 96% | 15 min | 80% |
| | | B | 2 m | 95% | 1.5 h | 100% |
| | | C | 1 m | 99% | 10 min | 90% |
| | 1,7 | A | 6 m | 96% | 45 min | 60% |
| | | B | 20 m | 96% | 20 h | 100% |
| | | C | 4 m | 99% | 30 min | 80% |
| | 1,8 | A | 150 m | No convergence | 48 h | 0% |
| | | B | - | - | - | - |
| | | C | 100 m | 93% | 5 h | 80% |
| | 5,5 | A | 1 m | 88% | 10 min | 80% |
| | | B | 1 m | 95% | 15 min | 98% |
| | | C | 600 k | 94% | 3 min | 90% |

**Table 2.** *Cont.*

| Number of Stages | (x,y)-IPUF | Security Evaluator | Training Size | Average Accuracy | Training Time | Success Rate |
|---|---|---|---|---|---|---|
| 64 bits | 6,6 | A | 5 m | 88% | 2 h | 50% |
| | | B | 5 m | 95% | 2.5 h | 75% |
| | | C | 3 m | 93% | 40 min | 80% |
| | 7,7 | A | 8 m | 87% | 2 h | 50% |
| | | B | 40 m | 95% | 17 h | 74% |
| | | C | 6 m | 92% | 1.5 h | 80% |
| | 8,8 | A | 150 m | No convergence | 48 h | 0% |
| | | B | 150 m | 95% | 1.5 weeks | 35% |
| | | C | 120 m | 91% | 30 h | 60% |

The results show that, first and foremost, the simple NN attacking method (method "A" in the table) succeeds in breaking the IPUF smaller than (1,8)-IPUFs or (8,8)-IPUFs within a given number of CRPs. However, for the IPUF with a larger size, the simple NN attacking method failed at 100 million training CRPs. In addition, the accuracy result coming from the method "A" is even lower than 90% when $x > 1$. Note that this number of required CRP is for a no-open interface.

As seen in Table 2, our pre-learning-based attack method significantly reduces the required number of CRP and required training time compared to the baseline method. For example, 300k CRPs are required to break 64-bit (1,5)-IPUFs by the subspace pre-learning method, while 1 million CRPs are required by the method "A" and 500 k CRPs are required by the method "B". In addition, the required number of CRP to break 64-bit (5,5)-IPUFs by the subspace pre-learning method is 600k, which is much smaller than the 1 million in the two comparison methods.

Moreover, the average accuracy of our proposed method is much higher than the average accuracy of method "A". In addition, for the IPUFs, the baseline method failed to break, (1,8)-IPUFs or (8,8)-IPUFs, and our subspace pre-learning-based attack method successfully breaks them with 100 million or 120 million. Compared with method "B", our method requires less CRP and significantly reduces the required training times, which is more efficient targeting the large-size IPUFs. For the IPUFs with a further larger size, like (9,9)-IPUFs, training set size and our Python code exceed the available memory of the computer node so that we stopped at (8,8)-IPUFs.

## 5. Conclusions

In this paper, we described a subspace pre-learning attack method for the IPUFs with an open interface. Unlike earlier studies on the vulnerabilities of IPUF either required prior knowledge of the IPUF architecture or only focused on plain challenge–response pairs without leveraging open-interface information, our method makes use of the IPUFs' open interface and does not require any prior knowledge of IPUF architecture. The method we employed could benefit from the pre-learning by subspace training datasets and speed up the attacking on the whole challenge space. When compared to previously published results, our study discovered a vulnerability in the IPUF with PUF circuit parameter values that were previously deemed secure. In particular, our method could break 64-bit (1,5)-IPUFs with 300 k CRPs and break 64-bit (5,5)-IPUFs with 600 k CRPs. We also could break 64-bit (1,8)-IPUFs with 100 m CRPs and break 64-bit (8,8)-IPUFs with 120 m CRPs. Both of these performances are superior to any earlier attack methods in the required number of CRPs or required training time, indicating that such IPUFs may be vulnerable if they have an interface that conforms to the way challenge–response data are accessed by our subspace pre-learning-based attack method. Importantly, this subspace pre-learning attack

method provides PUF manufacturers and IoT security application developers with valuable information for the protocol of current IPUF-based applications to mitigate potential risks. Our work does not fully break the IPUF for all conceivable sizes and complexity yet; however, future research could continue to explore the vulnerability of IPUFs with larger sizes and complexity by further utilizing the open interface of IPUFs.

**Author Contributions:** Methodology, G.L. and K.T.M.; software, G.L.; validation, G.L.; writing—original draft preparation, G.L.; writing—review and editing, G.L. and K.T.M. All authors have read and agreed to the published version of the manuscript.

**Funding:** This research received no external funding.

**Acknowledgments:** We thank the anonymous reviewers whose comments/suggestions helped improve and clarify this manuscript. In addition, the authors acknowledge the High Performance Computing Center (HPCC) at Texas Tech University for providing computational resources that have contributed to the research results reported within this paper. URL: http://www.hpcc.ttu.edu (accessed on 26 March 2021).

**Conflicts of Interest:** The authors declare no conflict of interest.

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
