# Peer review of "A Subspace Pre-Learning Strategy to Break the Interpose PUF"

_electronics, doi:10.3390/electronics11071049_

Round 1

Reviewer 1 Report

Physical Unclonable Functions (PUFs) are promising security primitives for resource constrained IoT devices. In this paper, the authors described a subspace pre-learning attack method for the IPUFs with an open interface. The topic is interesting. Some minor comments are given as follows:

  1. The main contributions should be further highlighted at the end of introduction by comparing with existing works.
  2. The results reveal vulnerability in IPUFs. What is the countermeasure?
  3. More description of Fig. 2 is welcome.
  4. How to determine the parameters of the NN attack method?

Author Response

We are very grateful for your comments on the manuscript. According to your advice, we amended the relevant part of the manuscript. The revised portion is marked in red in the paper. Some of your questions were answered below.

Point 1: The main contributions should be further highlighted at the end of introduction by comparing with existing works.

Response 1: Thank you for this very insightful comment. We have reworded several sentences to the Introduction section to further highlight the contributions by comparing them with existing works.

1- “For example, our method only needs 300k CRPs to break (1,5)-IPUFs, while 1m CRPs were needed in previous research; our method needs 1.5 hrs to break (1,7)-IPUFs, while 17 hrs were needed in previous research; etc.” (Line 53-55)

2- “Additionally, this attack method is capable of breaking (1,8)-IPUFs and (8,8)-IPUFs without prior knowledge about the position of the interpose bit, while previous research could only break (1,7)-IPUFs and (7,7)-IPUFs.” (Line 57-58)

Point 2: The results reveal vulnerability in IPUFs. What is the countermeasure?

Response 2: We are very grateful for your comments on the manuscript. That is a good question. The best countermeasure of reliability-based attack as we mentioned in Section 3 “When the IPUF was proposed and claimed to be immune to reliability attacks, the assumption was that IPUFs do not require a lockdown protocol to prevent open interfaces from being used by reliability attacks, and freely queried CRPs can be utilized for attacking IPUFs.”  (Line 116-119)

However, this countermeasure at the protocol level is costly. The PUF designer should revise the current protocol to reduce the number of allowed CRPs in lifespan to prevent attackers from gathering enough CRPs. We also added a sentence ” Importantly, this subspace pre-learning attack method provides PUF manufacturers and IoT security application developers valuable information for the protocol of current IPUF-based applications to mitigate potential risks.” to the Conclusion part to clarify this. (Line 231-233)

For another, there are some obfuscating techniques that can be used to mitigate the potential attacks at the protocol level. Preventing attacks at the protocol level makes them also costly and not efficient enough. In this paper, we prefer to focus on the basic attack resistance of PUFs, and so we will leave the obfuscating technique to other research.

Point 3: More description of Fig. 2 is welcome.

Response 3: Thanks for your comments, we have added a more detailed description under figure 2. Current description: “Illustration of an XOR-PUF with 2 arbiter PUF components and n bits of each stream, the final response is determined by the XORed result of two arbiter PUFs components.”

Point 4: How to determine the parameters of the NN attack method?

Response 4: Thank you for that excellent and insightful series of remarks. Determining the right parameters of the NN is based on studying the previous research that attacked the IPUF. In order to evaluate the performance of our method, we use exactly the same parameters of NN as the compared previous work as stated in section 4.2 “In our experiments, we chose the NN-based method used by Thapaliya et al. as the baseline method as well as the base learner method for the subspace pre-learning approach.” (line 174 to 175)

We also added a more detailed description  “The parameters of the NN attack method we used are exactly the same as the method used by Thapaliya et al to evaluate the performance.” after the statement to make it more clear. (line 178 to 180)

Reviewer 2 Report

-This paper describes a subspace pre-learning-based attack method to break Interpose Physical Unclonable Functions (IPUFs) that were considered as secure in an earlier study.
-The technical aspects of the research presented seem sound and detailed.
-Regarding the writing style and typos in the paper, I would suggest that the authors ask a native speaker of English colleague to proof read the paper. While most of the ideas and content can be understood, sometimes it is difficult to follow and that would also improve the presentation and quality of the paper.
-There is no need to include a header for Subsection 1.2 to indicate the organisation of the paper, that information can just be the last paragraph of Section 1.
-It would have been useful to mention applications of this kind of method in everyday life applications.
-The results obtained seem promising when compared to the other methods. I think you should mention future work at the end of the "Conclusions" section.

Author Response

We are very grateful for your comments on the manuscript. According to your advice, we amended the relevant part of the manuscript. The revised portion is marked in red in the paper. Some of your questions were answered below.

Point 1: Regarding the writing style and typos in the paper, I would suggest that the authors ask a native speaker of English colleague to proofread the paper. While most of the ideas and content can be understood, sometimes it is difficult to follow and that would also improve the presentation and quality of the paper.

Response 1: Thank you very much for your kind comments on our manuscript, the manuscript has been carefully revised by a native English speaker to improve grammar and readability.

Point 2: There is no need to include a header for Subsection 1.2 to indicate the organization of the paper, that information can just be the last paragraph of Section 1.

Response 2: Thanks for your comments, we have modified the organization by removing the header of subsections 1.1 and 1.2.

Point 3: It would have been useful to mention applications of this kind of method in everyday life applications.

Response 3: Thank you for this very insightful comment, we have added a sentence “Importantly, this subspace pre-learning attack method provides PUF manufacturers and IoT security application developers valuable information for the protocol of current IPUF-based applications to mitigate potential risks.” to clarify the application and real impact of our work. (Line 231-233)

Point 4: The results obtained seem promising when compared to the other methods. I think you should mention future work at the end of the "Conclusions" section.

Response 4: Thank you again for your positive and constructive comments and suggestions on our manuscript. We have added several sentences at the end of the Conclusion section “ Our work does not fully break the IPUF for all conceivable sizes and complexity yet, however, future research could continue to explore the vulnerability of IPUFs with larger sizes and complexity by further utilizing the open interface of IPUFs. ” to state the potential future work. (Line 233-236)
